# MaGA: Machine-Guided Amnesiac Unlearning through Target Feature Disentanglement

## Abstract

The security of training data has raised the "Right to be Forgotten" policy to protect the privacy of data providers, leading to an urgent need for effective Machine Unlearning. However, existing unlearning methods often face a trade-off dilemma between fully erasing the influence of target data and preserving the overall model capability. To address this, we first investigate the intrinsic characteristics of class concepts learned during model pretraining, revealing that these concepts are often entangled at the feature pattern level. Based upon this insight, we introduce Machine-Guided Amnesiac (MaGA), a novel unlearning framework to manipulate the unlearning process via leveraging Multi-modal Large Language Models to estimate conceptual similarities between features. These similarities are encoded in a transition matrix to assign suitable perturbing labels for re-alignment of target data to achieve unlearning. This facilitates effective unlearning, as it perturbs the concepts related to target instances, thus reducing undesired model disruption. Furthermore, we propose a Fragment-Absorb strategy to disentangle the influence of target concepts through a positive-negative feature noise pair. During unlearning, both feature noises are leveraged to impede target feature patterns while enhancing the remaining desired features. This promotes selective forgetting of target data influence, smoothing complete unlearning while mitigating the risks of under-unlearning or over-unlearning. Extensive experiments conducted across typical unlearning tasks and diverse datasets demonstrate that our approach outperforms existing baselines, effectively removing target data while preserving the model generalization on retained data.

## 1 Introduction

Machine Learning (ML), based on utilization of data resources, has been extensively explored in vast fields (Jordan & Mitchell, 2015; Lu & Weng, 2007; Nadkarni et al., 2011). However, studies (Jegorova et al., 2022; Fu et al., 2022) highlight the potential data security risks such as differential privacy (Dwork et al., 2014) and adversarial vulnerabilities (Steinhardt et al., 2017). Regulations (Voigt & Von dem Bussche, 2017; Bonta, 2022) have been introduced to address these concerns, where the "Right to be Forgotten" (Rosen, 2011) is underlined to enable deletion request from data providers to protect their privacy. To fulfill this target, an intuitive solution is to train a new model from scratch without target data. However, the substantial computation and time costs are simply unacceptable for practices. Thereby, the Machine Unlearning (MU) is proposed to tackle this problem, whose objectives are: 1) Remove the influence of specific data while maintaining the overall generalization; 2) Cost less time and computational resources than simply retraining.

Prior researchers mainly explore two types of unlearning strategies: model-centric and data-centric approaches. However, these approaches often face problems such as storage overhead and inaccurate unlearning. Another line of works (Graves et al., 2021), such as UNSIR (Tarun et al., 2023), crave to leverage feature noises to enhance forgetting of target data during fine-tuning. However, empirical evidence suggests a trade-off between effectively eliminating the influence of target data and maintaining the overall generalization capability.

To address this challenge, we first investigate in a question: *How does the unlearned data affect the retained model generalization?* Existing research (Serra et al., 2018) indicates that models acquire two levels of cognition during pretraining: **feature patterns** directly extracted from data, and

**semantic concepts** representing complex relations and combinations of different feature patterns, during pretraining. Prior study (Chang et al., 2024) interprets it as a mapping of features onto a higher-dimensional concepts space. This explains the interleaved influence that unlearned data poses on the retained model generalization, as different semantic concepts could share certain amounts of patterns, hereby named as associated features, while they each have unique features to distinguish from others. Figure 1 shows such an example. When unlearning the one of the concept, the associated features between two concepts are restrained, harnessing the generalization on the retained concept. Such interplay underscores the nature of the trade-off challenge. However, previous studies focus either on eliminating all related feature patterns of target concept (Tarun et al., 2023) or re-aligning the target concept with other retained concepts (Chen et al., 2023). The complex entanglement at the feature pattern level is neglected, leading to excessive or insufficient unlearning.

In this paper, we propose Machine-Guided Amnesiac (MaGA) as a framework to manipulate and enhance the unlearning process, as shown in Figure 2. Intuitively, MaGA aims to unlearn certain data by injecting misleading concepts to ensure learning a desired semantic gap from the target data. To achieve this, we leverage the guidance of zero-shot Multi-modal Large Language Models (MLLMs) to generate perturbing labels for finetuning. The similarities between different concepts are based on the understanding of prompted MLLMs. We store such a similarity in a transition matrix, which facilitates efficient inference of subsequent instance-based feature understanding. Furthermore, to address the problem of feature entanglement, we introduce the Fragment-Align strategy, which disentangles semantic concepts via a positive-negative feature noise pair. Specifically, the positive feature noise aims to align the target data representation with the

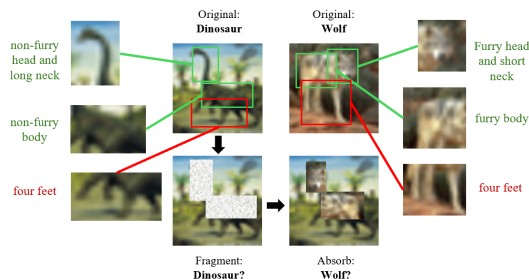

Figure 1: The entanglement among features from different concepts. Taking dinosaur and wolf as an example. They share similar features (marked as red) while each possesses unique features (green). The goal of our method is to distinctly process these two kinds of feature patterns.

semantics of the perturbing labels, while the negative feature noise disrupts the original representation associated with the true labels. Working together, these two complementary noises disentangle the target features from their original concepts and re-anchor them toward the perturbed concepts, enabling selective forgetting without harming overall generalization. Through substantial studies on three unlearning tasks conducted on a range of datasets, the efficacy and efficiency of MaGA as an unlearning method are rigorously validated. Our contributions in this paper can be summarized into:

- We demonstrate the intrinsic nature of model generalization with two levels of cognition: feature patterns and concepts. We further reveal the entangled interplay between different concepts, which leads to the trade-off challenge encountered by existing unlearning methods.

- We introduce the MaGA framework to manipulate the unlearning process by levearaging MLLM guidance. We further propose the Fragment-Align strategy to disentangle the influence of target data and solve the trade-off problem.

- We conduct comprehensive evaluations across three distinct unlearning tasks to assess and explore the effectiveness of MaGA.

## 2 RELATED WORKS

### 2.1 MACHINE UNLEARNING

Existing Machine Unlearning (MU) approaches can be categorized into two branches: model-centric and data-centric unlearning. Model-centric unlearning endeavors to repurpose knowledge from pre-trained models by selectively modifying or filtering their components or parameters, such as Sharded, Isolated, Sliced, and Aggregated (SISA) training (Bourtoule et al., 2021). Researchers (Yan et al., 2022; Zhou et al., 2022; Brophy & Lowd, 2021) continue to improve the performance through techniques such as data preprocessing. However, isolated training of model components will lead to generalization degradation and additional costs for initial training and storage. Model

pruning (Zhao et al., 2022; Liu et al., 2024b; Wang et al., 2022; Tanaka et al., 2020; Feldman, 2020; Stephenson et al., 2021), emerges to discard target data by selectively manipulating related crucial parameters (Yeom et al., 2021; Ma et al., 2021; Frankle & Carbin, 2018). Besides, the influence functions are typically approximated to estimate the essence of parameters to target data (Wu et al., 2022; Sekhari et al., 2021; Suriyakumar & Wilson, 2022; Foster et al., 2024; Golatkar et al., 2020a). Nevertheless, the risk of over-unlearning persists, as parameters important for forgetting data may also play a significant role for retained data (Chang et al., 2024). Data-centric unlearning aims to adjust pre-trained models through re-optimization and gradient updates (Neel et al., 2021; Cao et al., 2023; Graves et al., 2021; Shaik et al., 2024; Fan et al., 2023)to address forgetting requests. Golatkar et al. (2020a) introduce a scrubbing function during finetuning stage to align the unlearned model with the "gold model", which is continuously refined by subsequent research (Golatkar et al., 2021; 2020b; Shibata et al., 2021; Mehta et al., 2022; Tanno et al., 2022). DeltaGrad (Wu et al., 2020) and BAERASER (Liu et al., 2022) employ gradient updates using cached weight information during training to unlearn target data. Despite their achievements, methods of this kind rely on strong convexity assumptions, leading to unavoidable approximation errors. Alternative approaches have explored the utilization of feature noise (Tarun et al., 2023) and label noise (Chen et al., 2023) to diminish the generalization of forgetting data. Teacher-student framework (Chundawat et al., 2023; Zhang et al., 2023; Lin et al., 2023) is also employed to selectively distill knowledge, excluding forgetting data. While these methods provide intuitive solutions, they still face the inherent trade-offs problem of excessive or insufficient unlearning.

## 2.2 MULTI-MODAL LARGE LANGUAGE MODELS

The emergence of Transformer (Vaswani, 2017) facilitates the development of Large Language Models (LLM) (Brown, 2020; Floridi & Chiriatti, 2020; Touvron et al., 2023a;b). Incorporated with Vision Transformer (ViT) (Dosovitskiy, 2020), LLM is equipped with efficient multi-modal capabilities, known as Multi-modal Large Language Models (MLLM) (Li et al., 2022; 2023; Liu et al., 2024a; Dai et al., 2023; Ye et al., 2023). Within this framework, the representations of text and images are aligned through an intermediate structure. Building on this, MMICL (Zhao et al., 2023b) introduces a novel context training scheme to enable seamless insertion of image features into input text tokens, enhancing exceptional in-context learning capabilities. In this paper, we proposes to leverage the strengths of MLLMs to guide the unlearning process through a novel perturbing label assigning strategy. The in-context learning capabilities of MLLMs are utilized to estimate the similarities of feature patterns among different class concepts, which subsequently guarantees the disentanglement of influence of target data.

## 3 PRELIMINARIES

Given a dataset $\mathcal{D} = \{x_i, y_i\}_{i=1}^N$ where $y_i$ belongs to label space $\mathcal{Y} = \{1, \cdots, K\}$, the objective of machine learning is:

$$\theta = argmin_\theta \sum_{(x_i, y_i) \in \mathcal{D}} \mathcal{L}(f_\theta(x_i), y_i) \tag{1}$$

where $f_\theta$ is the DNN model parameterized by $\theta$, and $\mathcal{L}$ denotes the training loss function.

In machine unlearning, the goal is to remove the influence of a designated forget set $\mathcal{D}_f \subset \mathcal{D}$ from the pre-trained model $f_\theta$, while preserving its performance on the retain subset $\mathcal{D}_r = \mathcal{D} \setminus \mathcal{D}_f$. This process yields the unlearned model $f_{\theta'}$, where $\theta'$ represents the updated parameters. In this paper, we hypothesize that during the pre-training phase, the model learns a series of feature patterns and their corresponding mappings to semantic concepts, formalized as $\mathcal{P}_k \mapsto \mathcal{C}_k, k \in [1, K]$, where $\mathcal{P}_k = \{p_{k_1}, \cdots, p_{k_M}\}$ denotes the features of the semantic concept $\mathcal{C}_k$ from class $k$. For two distinct concepts $\mathcal{C}_i$ and $\mathcal{C}_j$, we define their feature intersection $\mathcal{P}_{ij}^{ass} = \mathcal{P}_i \cap \mathcal{P}_j$ as the **associated features**, while the **unique features** of concept $\mathcal{C}_i$ and $\mathcal{C}_j$ are defined as $\mathcal{P}_i^{uni} = \mathcal{P}_i \setminus \mathcal{P}_{ij}^{ass}$ and $\mathcal{P}_j^{uni} = \mathcal{P}_j \setminus \mathcal{P}_{ij}^{ass}$ other than the shared part.

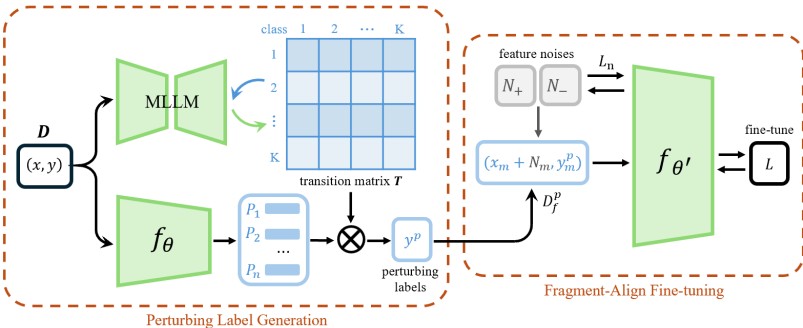

Figure 2: The proposed unlearning framework MaGA.

## 4 METHODOLOGY

In this section, we introduce MaGA for effective machine unlearning, which combines two components: 1) the MLLM guided label perturbation and 2) the Fragment-Align strategy. As shown in Figure 2, we first estimate inter-concept similarities using zero-shot MLLMs on a subset of training data, and cache them in a lightweight transition matrix. Further, based on the pretrained model's predictions, the transition matrix can assign proper label perturbation to achieve manipulation of concepts. For each forgetting instance, the pre-trained and cached feature noises are injected to disentangle their influence. Furthermore, we utilize MLLM guidance to quantify inter-concept distances to ensure effective label perturbation and identification of the class for positive noise, thereby preserving the retained data. In the following sections, we first demonstrate the process to generate perturbing labels in Section 4.1. Then, in Section 4.2, we carefully elucidate the Fragment-Align strategy for influence disentanglement and the fine-tuning method to unlearn target data.

### 4.1 PERTURBING LABELS GENERATION

Intuitively, to induce model unlearning, MaGA introduces semantically consistent but incorrect labels at the finetuning stage to replace the learned connections of target data to their correct labels with a meaningful but alternative relation to the retained concepts. To achieve this, we leverage MLLM to estimate the semantic consistency among concepts to assign suitable perturbing labels. To reduce the computation costs, we use a light-weight transition matrix to capture inter-concept similarities derived by the MLLM using a small subset from the training data. This matrix, encoding the knowledge of the MLLM, can be repetitively used for perturbing label assignment without further MLLM calls. In Appendix A, We provide the pseudo codes of our label assignment process for better understanding.

***Transition matrix estimation*** Specifically, let $q_w$ be the MLLM model parameterized by $w$. We randomly select $n$ exemplars from each class in the dataset $\mathcal{D}$ to construct the subset $\mathcal{D}_{ex}$. Then we prompt the MLLM model $q_w$ to estimate feature similarity of each instance in $\mathcal{D}_{ex}$ to all other different semantic concepts. For example, given a query image from class $k$ and $l \in [1, K]$, the prompt form is:

> Question: This image $<\text{IMG}_l>$ shows a photo of $<\text{label}_l>$, True or False? Answer: True;
> Question: This image $<\text{IMG}_p>$ shows a photo of $<\text{label}_l>$, True or False? Answer: False;
> Question: This image $<\text{IMG}_k^{query}>$ shows a photo of $<\text{label}_l>$, True or False? Answer:

In this way, the MLLM output is restricted to binary answers, i.e., True or False. Then, the feature similarity can be represented by softmax output logits.

To compute the feature similarity matrix among concepts, let $x_i$ be an instance from class $k$ in $\mathcal{D}_{ex}$, and $q_w(\cdot)$ be the MLLM output confidence. The feature similarity of $x_i$ with another concept $l$ can be represented as:

$$s_{kl} = q_w(x_i, l) \tag{2}$$

Then the feature similarity between two concepts labeled $k$ and $l$ can be approximated as:

$$\mathcal{S}_{kl} = \mathbb{E}(q_w(x_i, l)), (x_i, k) \in \mathcal{D}_{ex} \tag{3}$$

Subsequently, the transition possibilities of class $k$ with all other concepts can be calculated through: $\mathbf{t}_k = (\mathcal{S}_{k1}, \mathcal{S}_{k2}, \cdots, \mathcal{S}_{kK})^T / \sum_i^K \mathcal{S}_{ki}$. And the transition matrix $\mathbf{T}$, encoding all inter-conceptional similarities in $\mathcal{D}$, can be obtained via concatenation:

$$\mathbf{T} = (\mathbf{t}_1, \mathbf{t}_2, \cdots, \mathbf{t}_K) \tag{4}$$

Figure 3 illustrates the transition matrix of CIFAR-10 as an example. Although it reflects the proportion of overlapped features among different concepts, this is not adequate for label re-assignment as a given instance does not necessarily express all the features belonging to its concept. Simply applying $\mathbf{T}$ to assign perturbing labels will lead to biases followed by overfitting during unlearning. Therefore, it is essential to take the individual conditions of each instance into account.

***Assigning perturbing labels*** Given the transition matrix $\mathbf{T}$, the overlapping of an instance $x_i$ from class $k$ with other concepts at the feature level can be computed via:

$$\mathbf{R}_i = \mathbf{T} \cdot (\tilde{\mathbf{I}}_k \cdot f_\theta(x_i)) \tag{5}$$

where $\tilde{\mathbf{I}}_k$ denotes an identity matrix with the elements of $k$-th row elements set to zero to avoid the influence of the original concept. And $f_\theta(x_i)$ denotes the softmax output logits of the pretrained model.

Consequently, the perturbing label for an instance $x_i \in \mathcal{D}_f$ is obtained by ranking elements of $\mathbf{R}_i$:

$$y_i^p = \phi(\mathbf{R}_i, \tau) \tag{6}$$

Figure 3: The visualized transition matrix.

where $\phi(, \tau)$ denotes the process of selecting the top $\tau$ similar concept as the perturbing label. Here we set the parameter $\tau$ to control the distance between the target data and its re-assigned concept. As a perturbing concept that is too similar to the original one could fail to segregate the target data from the original embedding distribution, leading to insufficient unlearning. While the hindering of alignment with a too distant perturbing label will possibly cause model overfitting or representation collapse. In our experiments, we set $\tau = 0.3$ to ensure such a meaningful and alternative connection between target data and retained concepts. In Appendix E, we conduct detailed sensitivity studies to explain the selection of this parameter.

***Why not use model predictions for label assignment instead?*** Here, we demonstrate the necessity of MLLM guidance to assign perturbing labels, rather than using predictions from the pretrained model. Due to the overconfidence, the predicted probabilities of the pretrained model on a class of target data will be concentrated on a very few labels, neglecting the semantic content of individual instances. This poses a risk of leading the pretrained model to overfit to a simple output bias. Conversely, our mechanism, considering individual conditions of target data, guarantees balanced perturbing labels, as shown in Appendix C. These facilitate the model in adjusting its decision boundaries accordingly to different instances.

## 4.2 FRAGMENT-ALIGN STRATEGY AND FINETUING

In Section 1, we demonstrate that the key challenge of unlearning lies in feature entanglement: target data often share features with retained classes, making naive removal prone to over-forgetting. To address this, we propose the **Fragment-Align strategy**, which achieves semantic disentanglement through a pair of complementary feature noises. During fine-tuning, the **positive feature noise** $\mathcal{N}^{pos}$ enhances features aligned with the perturbing label, encouraging the target instance to re-anchor toward the new concept and mitigating distributional gaps. Conversely, the **negative feature noise** $\mathcal{N}^{neg}$ suppresses features tied to the original class, actively erasing its semantic association. Together, these two noises disentangle the target data from its original concept and realign it with the perturbed concept, facilitating the formation of new decision boundaries and enabling effective forgetting while preserving generalization on retained data.

***Feature noise generation*** Specifically, given an instance from target data, its ground-truth label $\mathcal{C}_{tar}$ and re-assigned perturbing label $\mathcal{C}_{per}$ correspond respectively to feature patterns $\mathcal{P}_{tar}$ and $\mathcal{P}_{per}$. We aim to eliminate the influence of the unique feature $\mathcal{P}_{tar}^{uni} = \mathcal{P}_{tar} \setminus \mathcal{P}_{tar|per}^{ass}$ while preserving shared features $\mathcal{P}_{tar|per}^{ass} = \mathcal{P}_{tar} \cap \mathcal{P}_{per}$ from unexpected disruption. Note that the feature noises can be pre-generated prior to the unlearning stages and reused across subsequent iterations to reduce computational and time costs. Here, we demonstrate the derivation of feature noises. Let $f_\theta$ be the pre-trained model, $(x_i, y_i)$ be an instance from forget data, $y_i^p$ be the perturbing label. The $\mathcal{N}$ is a randomly initialized matrix. The positive feature noise can be obtained through minimizing the loss function towards the perturbing label $y_i^p$ while keeping the pretrained model frozen:

$$\mathcal{N}_i^{pos} = \arg\min_{\mathcal{N}} \mathbb{E}(\mathcal{L}(f_\theta(\mathcal{N}), y_i^p) + \lambda \|w_{noise}\|) \tag{7}$$

Similarly, the negative feature noise can be obtained by changing the training direction towards the original label $y_i$ with a conversed loss function:

$$\mathcal{N}_i^{neg} = \arg\min_{\mathcal{N}} \mathbb{E}(-\mathcal{L}(f_\theta(\mathcal{N}), y_i) + \lambda \|w_{noise}\|) \tag{8}$$

Subsequently, incorporated with perturbing labels, the perturbed forget dataset is formulated as $\mathcal{D}_f^p = \{(x_i + \mathcal{N}_i, y_i^p) | x_i \in \mathcal{D}_f\}$, where $\mathcal{N}_i$ represents the weighted combination of positive and negative noises:

$$\mathcal{N}_i = \alpha \mathcal{N}_i^{pos} + (1 - \alpha) \mathcal{N}_i^{neg} \tag{9}$$

The constant $\alpha$ serves as a control parameter determining the weights of feature noises. At the fine-tuning stage, both the unique and associated feature patterns from concept $y_i$ are restrained by $\mathcal{N}_i^{neg}$. Simultaneously, the associated feature patterns shared with $y_i^p$ are restored by $\mathcal{N}_i^{pos}$, which helps construct alternative connections between perturbed data and the retained concept $\mathcal{C}_{per}$. Empirically, we set $\alpha = 0.7$ according to experiment results in Appendix E.

***Model re-optimization*** At the finetuning stage, we aim to change the model optimization direction shown in Equation 1 through a perturbed dataset $\mathcal{D}_f^p$, where representations of target data are pulled away from the original distribution and towards the retained data points. Specifically, the unlearning objective is achieved through updating model parameters where $\theta'$ denotes the updated parameters:

$$\theta' = \arg\min_{\theta'} \begin{cases} \sum_{(x_i, y_i^p) \in \mathcal{D}_f^p} \mathcal{L}(f_\theta(x_i + N_i), y_i^p) \\ \sum_{(x_i, y_i) \in \mathcal{D}_r} \mathcal{L}(f_\theta(x_i), y_i) \end{cases} \tag{10}$$

This process reshapes the decision boundaries with alternative embedding distributions. At the same time, the combination of positive and negative feature noises prevents the realignment from disrupting other data points via the disentanglement of target data features, where the positive feature noise additionally smooths the re-optimization. In Appendix F, we further explore the effects of these two components and their combination respectively through ablation studies.

## 5 EXPERIMENTS

In this section, we demonstrate experimental results on different unlearning tasks and datasets and compare the performance of MaGA with existing methods using a set of metrics. We randomly sample 10,000 instances from the retain dataset $\mathcal{D}_r$ along with the perturbed forget set $\mathcal{D}_f^p$ for fine-tuning using Eq. 10. We perform 3 epochs of unlearning for ResNet18, and 5 epochs for Vision Transformer (ViT). While increasing the number of unlearning epochs will further enhance performance, it comes at the expense of computational efficiency. Note that the metrics in tables are represented as percentages, and the boldings indicate superiority. For detailed hyper-parameter settings, we conduct sensitivity in Appendix E. To better understand the effects of perturbing labels and feature noises, we conduct ablation studies in the Appendix F by comparing MaGA with a randomly-selecting perturbing label strategy and a none feature noise framework. We also visualize the predictions of unlearned models compared with the "gold model" and baseline model to demonstrate the effects of unlearning in Appendix D. More results of class-wise and sub-class unlearning experiments on other target data are also present in Appendix G.

Table 1: Class-wise unlearning on CIFAR-100.

| model | class | metric | baseline | retrain | FT | UNSIR | AMNC | SSD | **MaGA** |
|-------|-------|--------|----------|---------|------|-------|-------|-------|----------|
| RN18 | RKT | $A_r$ | 76.30 | 76.19 | 65.43 | 73.83 | 73.59 | **75.86** | 75.75 |
|  |  | $A_f$ | 82.81 | 0.00 | **0.00** | 41.15 | **0.00** | **0.00** | **0.00** |
|  |  | MIA | 96.61 | 8.06 | 10.04 | 3.08 | 28.62 | 0.66 | **0.00** |
|  | MR | $A_r$ | 76.38 | 76.23 | 63.90 | 74.26 | 73.22 | 76.20 | **76.25** |
|  |  | $A_f$ | 82.03 | 0.00 | **0.00** | 8.07 | **0.00** | **0.00** | **0.00** |
|  |  | MIA | 95.65 | 6.01 | 12.22 | 1.68 | 46.06 | 0.25 | **0.00** |
| ViT | RKT | $A_r$ | 92.27 | 92.04 | 84.26 | 90.83 | 90.53 | 91.39 | **92.34** |
|  |  | $A_f$ | 93.14 | 0.00 | **0.00** | 24.57 | **0.00** | **0.00** | **0.00** |
|  |  | MIA | 84.88 | 6.29 | 16.00 | 11.43 | 1.06 | 6.62 | **0.00** |
|  | MR | $A_r$ | 92.20 | 92.18 | 84.15 | 89.95 | 89.95 | 91.78 | **92.33** |
|  |  | $A_f$ | 98.44 | 0.00 | **0.00** | 56.25 | **0.00** | **0.00** | **0.00** |
|  |  | MIA | 90.24 | 0.86 | 5.05 | 2.61 | 0.88 | 1.45 | **0.00** |

Table 2: Class-wise unlearning on CIFAR-20.

| model | class | metric | baseline | retrain | FT | UNSIR | AMNC | SSD | **MaGA** |
|-------|-------|--------|----------|---------|------|-------|-------|-------|----------|
| RN18 | veh2 | $A_r$ | 82.84 | 82.41 | 73.74 | 81.41 | **82.21** | 83.38 | 82.93 |
|  |  | $A_f$ | 84.99 | 0.00 | **0.00** | 58.82 | **0.00** | 22.37 | **0.00** |
|  |  | MIA | 87.72 | 14.24 | 40.84 | 45.68 | 7.84 | 5.72 | **0.08** |
|  | veg | $A_r$ | 82.59 | 82.24 | 72.56 | 81.23 | 81.66 | 82.64 | **82.53** |
|  |  | $A_f$ | 88.94 | 0.00 | **0.00** | 70.24 | **0.00** | 45.34 | **0.00** |
|  |  | MIA | 93.28 | 9.24 | 29.16 | 43.27 | 3.04 | 2.08 | **0.00** |
| ViT | veh2 | $A_r$ | 96.08 | 95.35 | 85.13 | 93.83 | 94.20 | 90.26 | **95.82** |
|  |  | $A_f$ | 94.35 | 0.00 | **0.00** | 67.29 | **0.00** | **0.00** | **0.00** |
|  |  | MIA | 80.96 | 20.36 | 22.00 | 49.96 | 1.26 | 9.88 | **0.00** |
|  | veg | $A_r$ | 95.88 | 94.95 | 88.52 | 93.75 | 93.31 | **95.61** | 95.64 |
|  |  | $A_f$ | 97.67 | 0.00 | 0.59 | 86.57 | **0.00** | **0.00** | **0.00** |
|  |  | MIA | 91.48 | 4.16 | 14.44 | 63.6 | 1.05 | 1.44 | **0.00** |

## 5.1 EXPERIMENTAL SETUP

**Datasets:** Following Foster et al. (2024), we evaluate our proposed method for image classification models using CIFAR-10, CIFAR-100 (Krizhevsky et al., 2009), CIFAR-20 (Xie et al., 2020).

**Models:** We leverage MMICL (Zhao et al., 2023a) as our MLLM zero-shot machine expert. We conduct unlearning experiments on two types of backbones: ResNet18 (He et al., 2016) and Vision Transformer (ViT) (Dosovitskiy, 2020). Models are trained using Adam optimizer (Diederik, 2014) and a multi-step scheduler with the initial learning rate set to 0.1. In convenience of comparisons, we control the training of the baseline model to align with that of the previous studies. The pretraining and unlearning processes are carried out on NVIDIA RTX4090 and Intel Xeon platform. The memory usage is controlled within 30 GiB for MLLM inference and less for unlearning.

**Unlearning tasks:** Following previous studies (Foster et al., 2024; Chundawat et al., 2023; Tarun et al., 2023), we evaluate the effectiveness of MaGA across three distinct unlearning tasks, including: 1) Class-wise unlearning conducted on CIFAR-100 and CIFAR-20, which unlearns an entire superclass from the dataset. 2) Sub-class unlearning conducted on CIFAR-20, where a sub-class within a superclass is forgotten. 3) Random unlearning conducted on CIFAR-10 where a subset randomly selected from the original dataset is forgotten.

**Baselines:** Following previous researches (Foster et al., 2024; Graves et al., 2021), we compare our method with baselines including: "*baseline*": the original pre-trained model, "*retrain*": the gold model trained from scratch, "*FT*" which finetunes the pre-trained model for 5 epochs with only retain data, "*UNSIR*" (Tarun et al., 2023), "*teacher*" which denotes Bad Teacher (Chundawat et al., 2023), "*AMNC*" which denotes Amnesiac (Graves et al., 2021), and "*SSD*" (Foster et al., 2024).

Table 3: Sub-class unlearning on CIFAR-20.

| model | sublass | metric | baseline | retrain | FT | UNSIR | AMNC | SSD | **MaGA** |
|-------|---------|--------|----------|---------|-----|-------|------|-----|------|
| RN18 | RKT | $A_r$ | 82.93 | 83.55 | 73.57 | 81.50 | 81.89 | 82.36 | **82.44** |
| | | $A_f$ | 81.51 | 1.39 | 15.36 | 60.85 | **0.00** | 5.90 | 4.51 |
| | | MIA | 84.43 | 22.40 | 12.29 | 43.6 | 9.41 | 5.85 | **4.80** |
| | sea | $A_r$ | 82.79 | 82.95 | 85.85 | 81.14 | 82.06 | 82.41 | **82.80** |
| | | $A_f$ | 97.66 | 77.34 | **76.14** | 95.49 | 35.76 | 86.02 | 84.38 |
| | | MIA | 87.10 | 56.67 | 66.05 | 86.60 | **4.00** | 54.65 | 66.68 |
| ViT | RKT | $A_r$ | 96.01 | 96.10 | 88.61 | 93.59 | 92.39 | 96.18 | **96.05** |
| | | $A_f$ | 94.70 | 2.83 | 5.12 | 66.75 | 0.78 | 22.83 | **1.10** |
| | | MIA | 85.80 | 7.44 | 19.41 | 15.47 | 0.86 | 3.45 | **0.00** |
| | sea | $A_r$ | 95.97 | 95.55 | 88.47 | 94.28 | 93.82 | 95.96 | **96.05** |
| | | $A_f$ | 98.44 | 74.22 | 81.68 | 89.15 | 17.97 | 97.66 | **12.76** |
| | | MIA | 89.26 | 41.43 | 43.27 | 69.20 | 0.22 | 86.49 | **0.20** |

Table 4: Random unlearning on CIFAR-10.

| model | metric | baseline | retrain | FT | teacher | AMNC | SSD | **MaGA** |
|-------|--------|----------|---------|-----|---------|------|-----|------|
| RN18 | $A_r$ | 95.31 | 92.42 | 87.86 | 90.68 | 90.63 | 91.38 | **93.04** |
| | $A_f$ | 94.03 | 94.40 | 88.77 | **93.13** | 55.35 | 95.88 | 85.59 |
| | MIA | 76.53 | 74.42 | 71.70 | 47.08 | **17.89** | 76.50 | 34.04 |
| ViT | $A_r$ | 98.66 | 98.75 | 97.64 | 98.17 | 98.15 | 98.63 | **98.81** |
| | $A_f$ | 99.71 | 99.41 | 98.24 | 93.52 | 74.88 | **99.39** | 96.27 |
| | MIA | 88.66 | 91.67 | 86.83 | 26.80 | **6.40** | 88.22 | 29.18 |

**Evaluation metrics:** Following Chundawat et al. (2023), we use: 1) Accuracy on forget and retain set, denoted as $A_f$ and $A_r$ respectively; 2) Membership Inference Attack (MIA) score (Shokri et al., 2017) as evaluating metrics. Considering the Streisand effect (Chundawat et al., 2023), we demonstrate that the unlearned model that aligns close to the gold model on accuracy is "well-unlearned". Meanwhile, lower MIA probabilities indicate better security in adversary attacks.

## 5.2 MAIN IMPLEMENTATION RESULTS

**Class-wise unlearning:** Experiments are conducted on CIFAR-100 and CIFAR-20 using ResNet18 and Vision Transformer as classification backbones. For CIFAR-100, we designate two classes: *rocket* (denoted as "RKT") and *mushroom* (denoted as "MR"). The results are shown in Table 1. Metrics demonstrate that our method successfully aligns its performance with the "gold model" denoted as "retrain". Especially when class-wisely unlearning *rocket* using ViT, MaGA narrows the gap with retrained model in retain accuracy by 0.35% compared to *SSD*. For CIFAR-20, we forget two superclasses: *Vehicle2* and *veg*. MaGA evidently lowers the retain accuracy compared with previous methods such as *SSD*, particularly decreasing over 40% on *vegetable* using ResNet18. Meanwhile, MaGA still maintains a competitive retain accuracy of 82.53%, which is closer to 82.24% of the "gold model". Thus, it is demonstrated that MaGA achieves a balance between complete unlearning and preservation of overall generalization. Crucially, MaGA significantly lowers the MIA risk compared to existing methods on most tasks, with 9.8% and 49% lower than *SSD* and *UNSIR* respectively on *veh2* from CIFAR-20 using ViT. This highlights the effectiveness of removing target information without being recognized by adversaries.

**Sub-class unlearning:** We perform unlearning on two sub-classes: *rocket* and *sea*, belonging to the super-classes "*Vehicle2*" and "*natural scenes*", respectively. The results are summarized in Table 3. The challenge of sub-class unlearning exists that the target sub-class shares feature patterns with fellows within the same super-class. Thus, when the features of the target sub-class are unlearned, the generalization capabilities of the entire super-class are compromised. This is evident in methods such as *UNSIR* and *Amnesiac*, which exhibit significant reductions in retain accuracy. In contrast, benefiting from the disentanglement of Fragment-Absorb strategy, the performance of MaGA on retained data can be well preserved and closely aligned with the "gold model". Taking sub-class

*rocket* with ViT backbone as an example, MaGA increases the retain accuracy by 2.51% and 3.71% compared to *UNSIR* and *Amnesiac* respectively. The superiority of MIA security of MaGA unlearning is also recurrently demonstrated among these cases, especially 86% and 69% lower than that of *SSD* and *UNSIR* on *sea* with ViT.

**Random unlearning:** We utilize the CIFAR-10 dataset for random unlearning, where a set of instances (100 in our experiments) is randomly selected as the target dataset to be forgotten. As shown in Table 4, random unlearning is inherently more difficult as the retrained gold model also persists a relatively high forget accuracy. In random unlearning, the forget set and retain set may contain samples from the same semantic class. Thus the unlearned model preserves generalization capability toward the class, making it possible to correctly classify forgotten samples despite their removal. More importantly, this indicates that, in such settings, the unlearning performance cannot be fully measured by accuracy alone. In this case, MaGA competitively reduces MIA scores compared to most baselines

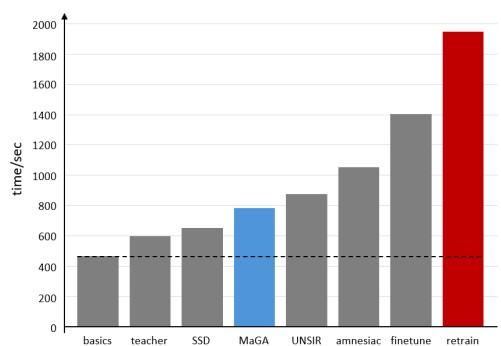

Figure 4: Time consumed for CIFAR-100 class-wise unlearning.

and the gold model, with 40% and 79% lower than *SSD* using ResNet18 and ViT respectively, which suggests that the model no longer remembers specific target instances, despite still leveraging general class-level knowledge. Simultaneously, it is still illustrated that MaGA behaves closely to the "gold model", with the overall generalization preserved after unlearning.

**Computation time:** We evaluate the time required to complete the unlearning process for each method, where the "basics" accounts for the time spent on dataset preparation, model loading, and metric computation. Using CIFAR-100 class-wise unlearning with the Vision Transformer (ViT) backbone as an example, MaGA reduces computational time by over 59% compared to full retraining, as illustrated in Figure 4.

# 6 CONCLUSION

**Contributions:** In this work, we introduce MaGA, a novel machine unlearning framework that effectively eliminate target data influence while maintaining overall model generalization through pioneering feature disentanglement and data realignment under the guidance of MLLMs. Extensive experiments across diverse datasets and forgetting tasks validate that MaGA consistently outperforms existing unlearning methods under most conditions, ensuring secured and effective unlearning. In future work, this framework can be extended to accommodate more complex unlearning scenarios and a broader range of pre-trained models.

**Limitations:** Due to the computation cost of utilizing MLLMs, MaGA is not the most time-efficient unlearning approaches. Nevertheless, this limitation can be mitigated through several strategies: 1) employing zero-shot inference from MLLMs solely as a form of knowledge guidance; 2) precomputing conceptual similarities and the transition matrix for a given dataset prior to the occurrence of any unlearning requests, and subsequently reusing the obtained transition matrix across all future unlearning tasks. Furthermore, the number of exemplars used in estimating the transition matrix can be flexibly adjusted to further reduce computational costs.

## REPRODUCIBILITY STATEMENT:

Our method can be reproduced by following the parameter settings provided in Appendix B, which allow replication of the experimental results. The code for the proposed approach will be released publicly in the future. It is worth noting that the algorithm involves certain stochasticity; therefore, results obtained using our code may exhibit slight variations compared to those reported in the paper, but they remain within an acceptable range.

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

# APPENDIX

## A  ALGORITHM OF PROPOSED MaGA

In Section 4, we propose MaGA as our unlearning method. MaGA employs zero-shot MLLM as machine experts to estimate the feature similarities between different concepts, represented by a transition matrix $\mathbf{T}$. This matrix is subsequently used to assign perturbing labels to the dataset, instructing the utilization of feature noises to disentangle the influence of the target concept. This approach ensures effective unlearning while preserving overall generalization. To better understand the workflow of perturbing label generation, we provide a detailed pseudo-code below.

---

**Algorithm 1** Transition matrix and perturbing labels.

---

**Input:** Instance $x_i$ of class $k$ from subset $\mathcal{D}_{ex}$ selected from training set $\mathcal{D}$, concepts $\mathcal{Y} = \{0, 1, \cdots, K-1\}$,
    MLLM $q_w$.
1: Let $q_w(x, y)$ represent the prompted MLLM output of image $x$ and concept $y$.
2: **for** $k \in \mathcal{Y}$ **do**
3:     **for** $l \in \mathcal{Y}$ **do**
4:         $\tilde{\mathcal{S}}_{kl} = \frac{1}{n}\sum_i^n [q_w(x_i, l)], (x_i, k) \in \mathcal{D}_{ex}$                                    ▷ Eq. 3
5:     **end for**
6:     $\mathbf{t}_k = (\tilde{\mathcal{S}}_{k0}, \tilde{\mathcal{S}}_{k1}, \cdots, \tilde{\mathcal{S}}_{kK-1})^T / \sum_i^{K-1} \tilde{\mathcal{S}}_{ki}$
7: **end for**
8: $\mathbf{T} = (\mathbf{t}_0, \mathbf{t}_1, \cdots, \mathbf{t}_{K-1})$
**Input:** Instance $(x_i, y_i)$ from forget set $\mathcal{D}_f$, transition matrix $\mathbf{T}$, class concepts $\mathcal{Y} = \{0, 1, \cdots, K-1\}$,
    pretrained model $p_\theta$, identity matrix $\mathbf{I}$ of size $[K, K]$, constant $\tau$.
9: Let $\phi(, \tau)$ denotes the process of selecting the index of the $\lfloor K\tau \rfloor$-th largest element.
10: **for** $(x_i, y_i) \in \mathcal{D}_f$ **do**
11:     $\tilde{\mathbf{I}}_{y_i} \leftarrow \mathbf{I}[j, y_i] = 0, j \in [0, K-1]$
12:     $\mathbf{R}_i = \mathbf{T} \cdot [\tilde{\mathbf{I}}_{y_i} \cdot p_\theta(x_i)]$                                       ▷ Eq. 5
13:     $y_i^p = \phi(\mathbf{R}_i, \tau)$
14: **end for**

---

## B  PARAMETER SETTINGS FOR EXPERIMENTS

During the unlearning process, different configurations are employed for the ResNet18 and Vision Transformer (ViT) backbones to reach a balance between effectiveness and efficiency. These settings are detailed in Table 5. The "noise size" refers to the batch size of feature noise samples extracted per class, which are randomly selected to be integrated with the target data. We at one time train a batch of feature noise for one class to enrich the variety. However, empirical results show that this will not affect the overall effectiveness of our method. Thus, the noise batch size can be decreased to save computation and time costs.

Table 5: Parameter setting for unlearning.

| Parameters | ResNet18 | ViT |
|---|---|---|
| exemplar $n$/class | 10 | 10 |
| batch size | 32 | 32 |
| noise size | 256 | 64 |
| noise lr | 0.1 | 0.1 |
| unlearn lr | 0.0003 | 0.00005 |
| rank $\tau$ | 0.3 | 0.3 |
| proportion $\alpha$ | 0.7 | 0.7 |
| number of instances for finetune | 10000 | 10000 |
| finetune iteration | 3 | 5 |

## C  DISTRIBUTION OF PERTURBING LABELS

In Section 4.1, we present a perturbing label assignment strategy grounded in the feature similarities between each instance in the forget set $\mathcal{D}_f$ and the retained concepts. This strategy leverages the

conceptual relationships encoded within the transition matrix $\mathbf{T}$, in conjunction with the predictions of $\mathcal{D}_f$ obtained from the unlearned model, to effectively quantify such feature-level affinities.

In contrast to unbalanced label assignment methods, which often mislead the model by systematically misclassifying target instances into a single erroneous class, our approach introduces diversity in the assignment of perturbing labels across instances. This variation mitigates the risk of inducing model bias and the Streisand effect during fine-tuning, by promoting a balanced and context-aware distribution of perturbing labels. Moreover, compared with complete random label assignment, our approach systematically considers the semantic compatibility between target instances and their assigned perturbing concepts. This compatibility is quantified through feature similarity, thereby guiding the reassignment process in a principled manner. Such alignment facilitates more effective unlearning, as target instances are more likely to be aligned with semantically related concepts, reducing unintended model disruption.

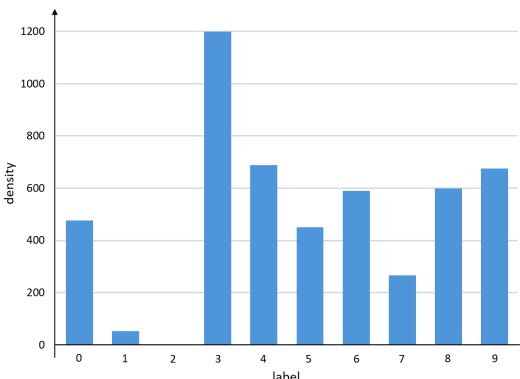

Figure 5: The distribution of perturbing labels for class-wise unlearning designating class 2 as target on CIFAR-10 with ResNet18.

Figure 5 visualizes the distribution of perturbing labels assigned to the forget dataset. This experiment is conducted on CIFAR-10, with class 2 designated as the target class, using ResNet18 as the backbone. While the observed disparities in perturbing label frequencies may partially reflect varying conceptual proximities to the target class, the overall distribution remains notably balanced, thereby validating the efficacy of our proposed assignment mechanism.

## D COMPARISON ON VISUALIZED MODEL PREDICTION BEHAVIOR AFTER UNLEARNING

The alignment of the unlearned model with the "gold model" is one of the most significant indicators when evaluating the unlearning performance, as is demonstrated in Section 5.1. In order to perceptually compare our proposed method with baselines, we utilize t-SNE to visualize the predictions of the unlearned model using different methods on CIFAR-10 test set. The results are shown in Figure 6.

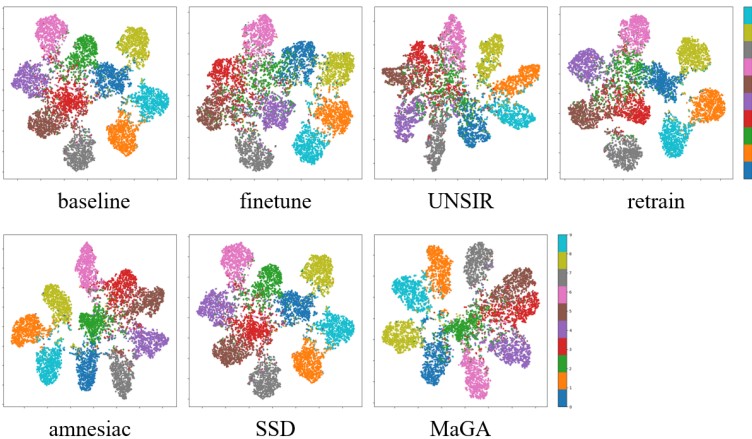

Figure 6: Visulized model prediction behavior comparison.

Each point in the figure represents an instance, while each color denotes a label (concept). Theoretically, for a model with better classification performance, the distributions of different concepts are better separated while that of the same concept should be more compact. However, in class-wise unlearning scenarios, we expect model generalization on the target class to be disrupted while preserving that on retained classes. Thereby, the distribution of unlearned concept, which is designated to be label 2 represented as green, is expected to be dispersed compared with the original pretrained

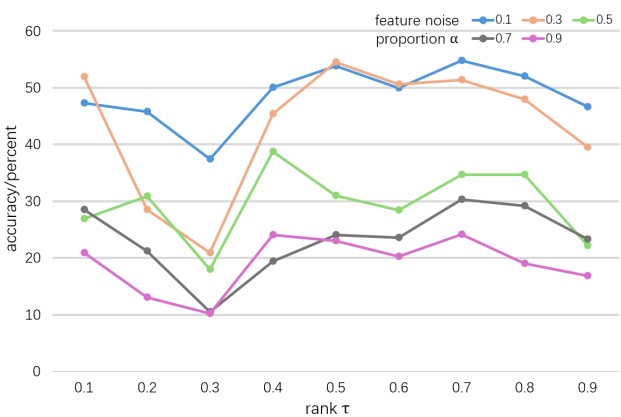

Figure 7: Performance on forget set under varying hyperparameters.

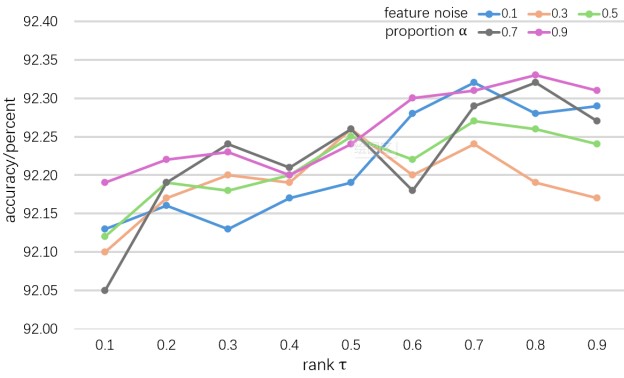

Figure 8: Performance on retain set under varying hyperparameters.

model (denoted as "baseline"). At the same time, the distributions of retain data are expected to maintain to preserve overall generalization. Comparing the prediction distributions of unlearned models, MaGA behaves more similar to the "gold model", with target instances split and captured by other well-separated retained concepts. This demonstrates the superiority of MaGA considering the closeness to the fully-retrained model.

## E    SENSITIVITY STUDIES

In our proposed method MaGA, two key hyperparameters, the rank constant $\tau$ and the noise proportion $\alpha$, play pivotal roles in the unlearning process. The rank constant $\tau$ regulates the feature similarity between the assigned perturbing labels and the corresponding target data. This is crucial, as an excessively high similarity between the reassigned and original concepts would result in the preservation of too many associated feature patterns, hindering adequate unlearning. Meanwhile, it is highlighted that the noise proportion $\alpha$ determines the balance between the positive feature noise encoded from the retaining data and the negative feature noise encoded from the forgetting data. As discussed in Section 4.2, this pair of feature noises poses different impacts on the influence disentanglement of target data during fine-tuning.

Here, to fully explore MaGA, we conduct sensitivity studies on these two hyperparameters. We record the class-wise performance of the unlearned model under varying settings using CIFAR-100 dataset. To more explicitly demonstrate the influence of different hyper-parameters on retain accuracy, we manually induce incomplete unlearning by setting the learning rate to 1e-5, which is much lower than our other experiments. We conduct 3 independent experiments for each hyperparameter setting. The averaged accuracy performance is calculated to reflect the general trend. Figure 7 and Figure 8 illustrate the classification accuracy on forget and retain dataset. It is shown that generally, a higher $\alpha$ represents lower accuracy in the retain set, enhancing the effects of positive feature noise on aligning

Table 6: Ablation studies on class-wise unlearning with CIFAR-100.

| model | class | Metric | baseline | retrain | SSD | RND | w.o.F.N | MaGA |
|---|---|---|---|---|---|---|---|---|
| RN18 | rocket | $Ar$ | 76.30 | 76.19 | 75.86 | **76.24** | 75.46 | 75.75 |
| | | $Af$ | 82.81 | **0.00** | **0.00** | 56.51 | 3.32 | **0.00** |
| | | MIA | 96.61 | 8.06 | 0.66 | 20.4 | **0.00** | **0.00** |
| | MR | $Ar$ | 76.38 | 76.16 | 76.20 | 75.98 | 76.07 | **76.25** |
| | | $Af$ | 82.03 | **0.00** | **0.00** | 57.12 | 4.55 | **0.00** |
| | | MIA | 95.65 | 5.61 | 0.25 | 10.21 | **0.00** | **0.00** |
| ViT | rocket | $Ar$ | 92.27 | 91.84 | 91.39 | **92.37** | 91.96 | 92.34 |
| | | $Af$ | 93.14 | **0.00** | **0.00** | 56.51 | 3.84 | **0.00** |
| | | MIA | 84.88 | 6.29 | 6.62 | 2.8 | 0.00 | **0.00** |
| | MR | $Ar$ | 92.20 | 92.18 | 91.78 | 92.07 | 91.79 | **92.33** |
| | | $Af$ | 98.44 | **0.00** | **0.00** | 72.66 | 10.17 | **0.00** |
| | | MIA | 90.24 | 0.86 | 1.45 | 1.4 | **0.00** | **0.00** |

perturbed instances with retain data. At the same time, the trend of retain accuracy under a changing rank $\tau$ is investigated. The prior hypothesis is verified that although we intuitively expect a more similar concept as the perturbing label, such a perturbing label with too many associated feature patterns will hinder the forgetting of target information. On the other hand, there is hardly clear regularity corresponding to $\alpha$ or $\tau$ due to the saturated classification performance on retain data. Thus, in experiments, a $\tau$ around 0.2 to 0.3 and an $\alpha$ around 0.7 to 0.9 are preferred.

## F ABLATION STUDIES

To deepen the understanding of the intrinsic properties of the MaGA framework and evaluate the impact of its key components: machine-guided perturbing labels and the following pair of feature noises. We conduct ablation studies using various combinations of these elements on class-wise unlearning tasks with CIFAR-100. The results are presented in Table 6, where 'RND' refers to assigning random labels to forget data instead of leveraging calculations of associated features, and 'w.o.F.N' represents fine-tuning exclusively with perturbing labels, omitting feature noises.

Compared with RND, the collaboration of MLLM guidance evidently facilitates the complete forgetting of target data, with over 50% reduction of retain accuracy and 20% decreased MIA on class rocket. It is attributed to the fine realignment effect of perturbing labels, which prevents the connections between retained feature patterns and target concepts. Without the MLLM guidance, the Fragment-Align strategy would not have functioned correctly, leading the model towards possibly the wrong tuning direction. Meanwhile, it is demonstrated that the addition of feature noises, including a pair of positive and negative feature noise, effectively disentangles and manipulates the generalization of the forget data. This guarantees a balance between complete unlearning of forgotten data and preservation of model generalization on retained data. For unlearning on class mushroom using ViT, feature noises help increase the retain accuracy by 0.54% while reducing forget accuracy by 10.17%.

## G SUPPLEMENTARY EXPERIMENTS

In Section 5, we demonstrate the effectiveness of MaGA in unlearning through a series of experiments, encompassing both class-wise and sub-class unlearning tasks on CIFAR-100 and CIFAR-20 datasets. To provide a more comprehensive evaluation of its performance, we extend the unlearning implementation to a larger number of classes (sub-classes).

**Additional results of class-wise unlearning** Table 7 presents results of unlearning across five different classes, where *DINO* denotes *dinosaur*. Compared to existing methods, MaGA-unlearned models exhibit a competitive alignment effect towards the retrained model. The retain accuracy of MaGA is notably maintained, showing an increase of 0.16% over SSD for the class *dinosaur*, which can be attributed to the disentanglement of retained and forgotten concepts. Additionally, it is observed that the MIA (Membership Inference Attack) risk of MaGA-unlearned models is significantly reduced. For example, the class *sea*, which presents a challenge even for the retrained model, shows a 0.4% reduction in MIA, bottoming 1.20% with MaGA, which is substantially lower

Table 7: Additional results of class-wise unlearning on CIFAR-100.

| model | class | metric | baseline | retrain | FT | UNSIR | AMNC | SSD | MaGA |
|-------|-------|--------|----------|---------|-----|-------|------|-----|------|
| RN18 | baby | $A_r$ | 76.54 | 75.71 | 64.64 | 71.86 | 73.53 | 76.64 | **76.70** |
| | | $A_f$ | 67.80 | **0.00** | **0.00** | **0.00** | **0.00** | **0.00** | **0.00** |
| | | MIA | 96.27 | 3.44 | 19.30 | 18.46 | 54.81 | **0.00** | **0.00** |
| | lamp | $A_r$ | 76.50 | 75.33 | 64.74 | 71.64 | 73.32 | **76.46** | 76.11 |
| | | $A_f$ | 69.10 | **0.00** | **0.00** | **0.00** | **0.00** | **0.00** | **0.00** |
| | | MIA | 96.20 | 0.63 | 10.74 | 8.68 | 51.89 | **0.00** | **0.00** |
| | sea | $A_r$ | 76.35 | 73.16 | 64.00 | 71.14 | 73.45 | **75.11** | 73.36 |
| | | $A_f$ | 83.68 | **0.00** | **0.00** | **0.00** | **0.00** | **0.00** | **0.00** |
| | | MIA | 9.42 | 4.85 | 34.41 | 36.67 | 27.66 | 1.60 | **1.20** |
| | DINO | $A_r$ | 76.39 | 74.79 | 63.54 | 71.63 | 73.43 | 76.39 | **76.55** |
| | | $A_f$ | 78.13 | **0.00** | **0.00** | **0.00** | **0.00** | **0.00** | **0.00** |
| | | MIA | 98.20 | **0.00** | 12.24 | 5.87 | 36.86 | **0.00** | **0.00** |
| | wolf | $A_r$ | 76.38 | 76.28 | 63.45 | 71.75 | 72.64 | 76.26 | **76.30** |
| | | $A_f$ | 79.51 | **0.00** | **0.00** | **0.00** | **0.00** | **0.00** | **0.00** |
| | | MIA | 97.40 | 0.49 | 13.62 | 13.84 | 43.45 | **0.00** | **0.00** |

Table 8: Additional results of sub-class unlearning on CIFAR-20.

| model | subclass | metric | baseline | retrain | FT | UNSIR | AMNC | SSD | MaGA |
|-------|----------|--------|----------|---------|-----|-------|------|-----|------|
| RN18 | beetle | $A_r$ | 82.63 | 82.35 | 73.23 | 82.00 | 82.20 | 80.10 | **82.37** |
| | | $A_f$ | 82.64 | 67.71 | 72.05 | 76.13 | 40.97 | 0.78 | **70.40** |
| | | MIA | 88.43 | 20.26 | 58.64 | 71.00 | 9.40 | 8.81 | **6.26** |
| | snail | $A_r$ | 83.00 | 81.97 | 75.49 | 80.98 | 82.80 | **82.77** | 83.10 |
| | | $A_f$ | 69.88 | 37.59 | 16.32 | 55.30 | 9.90 | 6.68 | **50.27** |
| | | MIA | 84.50 | 9.47 | 16.57 | 47.22 | 10.05 | 4.80 | **4.33** |
| | whale | $A_r$ | 82.90 | 82.09 | 74.81 | 81.66 | **82.08** | 82.67 | 82.74 |
| | | $A_f$ | 77.08 | 67.27 | 61.37 | 79.08 | 20.92 | 72.57 | **72.55** |
| | | MIA | 85.86 | 36.47 | 54.39 | 77.80 | **3.87** | 61.42 | 26.71 |
| | fox | $A_r$ | 82.96 | 82.40 | 71.50 | 82.06 | 81.98 | 79.90 | **82.32** |
| | | $A_f$ | 72.40 | 6.08 | 17.71 | 56.68 | **4.34** | 0.00 | 13.12 |
| | | MIA | 82.46 | 5.60 | 15.66 | 35.43 | 14.61 | 21.80 | **3.70** |
| | SCP | $A_r$ | 82.80 | 82.08 | 74.00 | 81.39 | 82.35 | 81.62 | **82.29** |
| | | $A_f$ | 90.36 | 67.53 | 61.02 | 82.12 | 5.12 | 7.46 | **69.14** |
| | | MIA | 91.49 | 16.20 | 33.00 | 56.02 | 4.28 | 5.45 | **3.94** |

than that of the retrained model. This further validates the security of the unlearning method proposed in our work.

**Additional results of sub-class unlearning** Table 8 presents the results of sub-classes as unlearning targets, where *SCP* denotes *skyscraper*. While certain sub-classes, such as *whale* and *skyscraper*, present more challenges in the unlearning process, MaGA still outperforms existing methods by maintaining a high alignment with the accuracy of the retrained models. For sub-classes *beetle*, *snail*, *whale*, and *skyscraper*, MaGA sticks closely to the retrained models in terms of forgetting accuracy. Additionally, it significantly reduces MIA risks across all conditions. These results further demonstrate the effectiveness of our proposed method in sub-class unlearning tasks.

