# OpenReview forum: "MaGA: Machine-Guided Amnesiac Unlearning through Target Feature Disentanglement"
_ICLR.cc/2026/Conference — Submitted to ICLR 2026_

### Official Review · Reviewer_Xuxu · 2025-10-27

**Soundness:** 1
**Presentation:** 1
**Contribution:** 2
**Rating:** 2
**Confidence:** 5

**Summary:**

This paper proposes MaGA, a new machine unlearning framework that integrates Multi-modal Large Language Models (MLLMs) to guide the unlearning process. The method introduces two key ideas: 1. A transition matrix constructed from MLLM-estimated conceptual similarities to generate perturbing labels for unlearning; 2. A Fragment-Align strategy that injects positive and negative feature noises to disentangle target features and preserve generalization. Experiments are conducted on CIFAR-10, CIFAR-20, and CIFAR-100 across class-wise, sub-class, and random unlearning tasks. Results suggest that MaGA achieves competitive performance compared with existing methods, offering improved unlearning effectiveness (lower MIA) and retained accuracy close to retrained “gold” models.

**Strengths:**

1. Novel conceptual direction: Leveraging MLLMs for semantic similarity estimation in unlearning is an interesting and creative approach that bridges recent advances in multimodal understanding and privacy-oriented model editing.
2. Relatively well-designed experiments: Includes comparisons to multiple baselines, ablation studies, sensitivity analysis, and visualization (t-SNE), showing careful experimental design.

**Weaknesses:**

1. High computational overhead and scalability concerns. MaGA heavily depends on MLLM inference and feature noise generation. The method is only demonstrated on small datasets (CIFARs), and it remains unclear how it scales to larger datasets (e.g., ImageNet) or higher-dimensional data. The claimed precomputation trick for the transition matrix is not empirically verified.
2. Lack of theoretical justification. The notion of “feature disentanglement” and the claimed semantic alignment between perturbing and target labels are conceptually appealing but lack theoretical analysis or measurable metrics. The method’s convergence or stability is not discussed.
3. Limited experimental diversity. All experiments are on small-scale vision datasets. The method’s generality to real-world or cross-domain settings (e.g., text, multimodal, federated unlearning) is untested.
4. Relation to prior art unclear. The proposed Fragment-Align approach is conceptually close to existing feature-noise-based unlearning methods. The paper does not sufficiently differentiate MaGA in terms of objective function or theoretical insight.
5. Efficiency not convincingly addressed. Although the paper claims 59% time savings compared to full retraining, the absolute time cost of MLLM queries is omitted, and the comparison seems unfair because the baseline retraining cost depends on dataset scale.
6. Poor writing. The author should improve the illustration through adjusting figure, table and more fluent expression.

**Questions:**

See the weaknesses.

---

> ### Author Response · Authors · 2025-11-19
> **Author Rebuttal P1**
>
> Thanks for your suggestions.
> - W1: Concerns about MLLM computation.
>   - As clarified in Sec. 4.1–4.2, MLLM inference is **not the computational bottleneck** , and is excluded from Fig. 4 for the following reasons:
>     - **One-time, zero-shot usage:** MLLM is queried only once per dataset to estimate inter-class semantic similarity. No training or fine-tuning is required, keeping the cost minimal
>     - **MLLM is not queried for each sample:** The transition matrix is computed using only 3–10 exemplar images per class (about 5 instances are typically sufficient) to cache the MLLM knowledge of semantic relations. Therefore, the computation cost scales with **the number of classes** in the dataset, not the dataset size, guaranteeing the scalability of MaGA.
>     - The estimation of the transition matrix is fully **decoupled from the training or unlearning process**. It is computed once per dataset, **even before model training**, and reused for all subsequent unlearning tasks. Its cost is thus amortized and irrelevant to per-task efficiency.
>   - Feature noise generation is **lightweight** and also **can be reused**. They are small-scale vectors (1-5 per batch will be enough for feature diversity) trained for only 3-5 epochs on a frozen backbone in a parallel way, consistent with the finding of [1].
>   - [1] Tarun A K, Chundawat V S, Mandal M, et al. Fast yet effective machine unlearning[J]. IEEE Transactions on Neural Networks and Learning Systems, 2023, 35(9): 13046-13055.
> - W5: Demonstration of efficiency according to Figure 4.
>   - We respectfully disagree that the efficiency comparison is unfair. Our evaluation follows standard MU practice:
>     - Including MLLM’s one-time precomputation is unnecessary for per-task efficiency evaluation, as explained above. The MLLM is queried exactly once per dataset and reused thereafter. Thus, counting this cost into every unlearning run would **artificially inflate the runtime and misrepresent real deployment settings**.
>     - **Retraining-from-scratch is the canonical MU baseline**, since it represents the naïve solution MU aims to surpass. All methods are evaluated under an identical dataset scale and protocols, ensuring fairness.
>     - Beyond retraining, MaGA is also more efficient than baselines such as UNSIR and AMNESIAC, achieving competitive $A_f$ and $A_r$, and lower MIA risk with faster runtime. This demonstrates MaGA’s efficiency advantage beyond the retrain comparison alone.
>   - The reported 59% reduction reflects true per-task computational savings; we will make this clearer in revision.
> - W3: About experiment diversity.
>   - Our work is positioned within the scope of **classification-model unlearning**, consistent with dominant prior studies (Sec. 2). The reviewer’s suggested domains—text, multimodal, federated unlearning—fall into **distinct research scope** with different problem formulations and evaluation regimes. We would appreciate relevant references for us to follow if such cross-domain baselines exist.
>   - Within the classification setting, MaGA is theoretically motivated and empirically validated. Following another reviewer’s insight, we also evaluate MaGA on generative unlearning, following [2]. We compare MaGA against ESD and SalUn on CIFAR-10 concept unlearning using a DDPM with classifier-free guidance. Two target classes are selected, and performance is evaluated using the forgetting quality (UA) from an external classifier and the FID estimated on retained classes. MaGA remains competitive and closely aligned with the retrained model, demonstrating transferability beyond classification.
> | class	| metric	| retrain	| ESD	|  SalUn	| MaGA  |
> |--------|--------|-------|-------|-------|-------|
> | Airplane | UA $\uparrow$		| 100.00	| 95.10	| 94.65	| 94.52	|
> | Airplane | FID $\downarrow$	| 12.86	| 18.25	| 12.03	| 12.41	|
> | Cat		| UA $\uparrow$		| 100.00	| 93.58	| 93.44	| 93.76	|
> | Cat 	| FID $\downarrow$	| 13.10	| 16.85	| 11.28	| 12.33	|
>   - [2] SalUn: Empowering Machine Unlearning via Gradient-based Weight Saliency in Both Image Classification and Generation.

---

> ### Author Response · Authors · 2025-11-19
> **Author Rebuttal P2**
>
> - W2: Theoretical justification of feature disentanglement in MaGA.
>   - Feature disentanglement in MaGA arises from the paired positive–negative noises in Fragment-Align (Sec. 4.2).
>     - With the pretrained model frozen, the noise $N$ can only reduce the loss when its perturbation aligns with the model’s class-level latent semantics (Eq. 7) and increase it when misaligned (Eq. 8), which in fact **modulate latent semantic directions** already encoded in the pretrained model, ensuring it learns semantic directions rather than arbitrary perturbations.
>     - Prior work [1] shows negative noise can overwrite class-relevant weights. We extend this principle using positive noise to inject semantics of the perturbing class. Together, they enable explicit disentanglement and semantic realignment guided by MLLM-derived labels by simultaneously weakening the original feature-concept relations and strengthening the alternative concept association.
>   - From an **optimization viewpoint**, MaGA can be viewed as reshaping the pretrained feature space: negative noise disrupts the target distribution, while positive noise reassigns representations to new semantic regions, enabling realignment without full retraining. And this optimization direction is guided by the MLLMs’ knowledge through perturbing labels.
>   - Empirically, Appendix F provides measurable evidence:
>     - The “w.o.F.N” variant (without feature noises) fails to balance forgetting and retention compared to MaGA, indicating the necessity of feature disentanglement.
>     - Additionally, we add extended ablations below comparing “w.P.F”(with positive feature), “w.N.F”(with negative feature), and MaGA. It is observed that neither “w.P.F.” nor “w.N.F” variants suffice. Specifically, lack of positive features(“w.N.F”) leads to degradation in preserving retained generalization, while “w.P.F” cannot fully unlearn target information. Only the full Fragment-Align strategy achieves the closest performance to the retrained model.
> | class	| metric	| retrain	| w.o.F.N	| w.P.F	| w.N.F	| MaGA	|
> |--------|--------|-------|-------|-------|-------|-------|
> | rocket	| $A_r$	| 76.19	| 75.46	| 75.91	| 74.63	| 75.75	|
> | rocket	| $A_f$	| 0.00	| 3.32	| 10.35	| 3.44	| 0.00	|
> | rocket	| MIA	| 8.06	| 0.00	| 8.40	| 0.00	| 0.00	|
> | MR    | $A_r$	| 76.23	| 76.07	| 76.22	| 74.78	| 76.25	|
> | MR    | $A_f$	| 0.00	| 4.55	| 9.50	| 0.00	| 0.00	|
> | MR    | MIA	| 6.01	| 0.00	| 10.20	| 1.68	| 0.00	|
> - W4: Re-demonstration of the relation between MaGA and the prior work UNSIR.
>   - We respectfully disagree that Fragment-Align is conceptually close to prior feature-noise methods such as UNSIR [1]. Although both optimize noise, their intentions and mechanisms differ fundamentally:
>     - **Different conceptual intuition:** As discussed in Sec. 2, UNSIR employs an error-maximizing noise solely to damage the model’s reliance on target-class features. In contrast, MaGA is driven by an MLLM-guided semantic signal: perturbing labels provides concept-level supervision that enables Fragment-Align to explicitly disentangle target-specific and shared semantic features. Instead of merely destroying target features, it restructures the relationship between features and retained concepts.
>     - **Different roles of noise:** MaGA adopts **paired positive–negative noises**: negative noise suppresses target-unique semantics, while positive noise aligns target data toward alternative concepts. UNSIR’s negative noise aligns with pushing representations away from a class subspace, while MaGA extends this principle by adding positive noise that pulls representations toward an alternative semantic direction determined by the perturbing labels. This enables the **model re-optimization** that reconstructs the latent space and decision boundary after the disentanglement. However, UNSIR performs no such semantic realignment.
>     - **Empirical results support the distinction:** Experiments in Sec. 4 show that MaGA consistently achieves lower forgetting accuracy and higher retention accuracy than UNSIR, effectively resolving the $A_r$-$A_f$ trade-off. These improvements stem directly from Fragment-Align’s disentanglement and semantic realignment capabilities, which are absent from UNSIR.
> - W6: Could you kindly specify the problems so that we can improve the clarity of the paper?

---

> ### Author Response · Authors · 2025-11-25
> **Further Discussion**
>
> Dear Reviewer Xuxu:
>
> Thank you for taking your valuable time to review our paper. Your comments were very helpful.
>
> We have made our most efforts to address your concerns, including the computational costs of MLLM, the scope of the research, the theoretical justification of feature disentanglement, and the relation/differences between MaGA and the prior work. We also add experiments on generative tasks to demonstrate the extendibility of MaGA.
>
> Thank you again for your feedback. We look forward to any further comments you may have.
>
> Best,
>
> Authors.

---

> > ### Comment · Reviewer_Xuxu · 2025-11-27
> >
> > Thank you for your feedback and detailed explanation. I have raised my score to 4. However, I still believe that this version hasn't achieved the bar of ICLR. I would suggest that you should make a more systematic and detailed revision to this paper and resubmit it.

---

> > > ### Author Response · Authors · 2025-11-28
> > >
> > > Thanks for your response!
> > >
> > > Here we would like to reiterate the key contributions of our work, including:
> > > - We propose a **novel concept-aware unlearning algorithm** that leverages MLLM-guided re-optimization with a feature-disentanglement mechanism, effectively addressing the trade-off between over-unlearning and insufficient unlearning.
> > > - We conduct **extensive experiments** across diverse unlearning tasks and datasets to thoroughly evaluate and validate the proposed method.
> > > - We provide a **clear and detailed explanation** of the algorithmic procedure, which reviewers noted as “inspiring for follow-up research” (R2) and “easy to follow” (R1).
> > >
> > > We will further revise the paper to enhance clarity. Thank you again for your valuable time. Please let us know if you have any remaining concerns.

---

### Official Review · Reviewer_pJCC · 2025-10-31

**Soundness:** 3
**Presentation:** 2
**Contribution:** 2
**Rating:** 4
**Confidence:** 4

**Summary:**

The key idea of this work is to maintain a balance between effective forgetting of targeted concepts and reducing undesired model disruption. To achieve this, this work first constructs a Transition matrix that aligns perturbed labels by considering inter-conceptional similarities. Additionally, to balance over-unlearning and under-unlearning, this work introduces positive and negative perturbation noises, thereby maintaining retained knowledge while selectively forgetting the target concepts.

**Strengths:**

1) This paper explores the over-unlearning and under-unlearning, which a key challenge of machine unlearning. To address it, this work proposes a balanced mechanism that preserves retained knowledge while effectively forgetting targeted concepts.

2) The use of semantic concept modeling (inter-concept similarity via a transition matrix) is insightful and could inspire follow-up research on concept-aware unlearning.

**Weaknesses:**

1) The theoretical foundation behind the proposed method is unclear. In particular, the derivation of positive and negative noise terms in Equations (8) and (9) does not guarantee the generation of semantically meaningful concepts. Previous studies on model inversion and adversarial attacks have shown that generating interpretable concepts requires explicit constraints, which are not discussed or justified in this paper.

2) The experiments focus mainly on classification tasks, with no evaluation on multi-modal or generative tasks. Since the method operates at the concept level, it would be particularly relevant and insightful to test it on conceptual generative models.

3) The paper does not clearly justify why a Multimodal Large Language Model is required to construct the Transition Matrix. It remains unclear whether this component is central to the proposed method. Moreover, the lack of ablation studies makes it difficult to assess its contribution and necessity.

4) The experimental results are not fully convincing. For example, Table 1 (class-wise unlearning on CIFAR-100 using ResNet-18) shows inconsistent improvements, and the use of bolded values is confusing. Furthermore, the evaluation omits comparison with important recent baselines such as SalUn [1].

[1] SalUn: Empowering Machine Unlearning via Gradient-based Weight Saliency in Both Image Classification and Generation

**Questions:**

1) How to ensure that the positive and negative noise vectors correspond to semantically meaningful concepts rather than arbitrary perturbations?

2) Is MLLM specifically used to build the Transition Matrix? Would a simpler embedding-based similarity approach (e.g., CLIP or Word2Vec) yield comparable results?

3) Could the approach generalize to generative models or LLMs?

---

> ### Author Response · Authors · 2025-11-19
> **Author Rebuttal P1**
>
> Thanks for your constructive comments.
> - W1: The theoretical demonstration of the proposed method.
>   - We appreciate the reviewer’s concern about the theoretical basis of the proposed positive and negative feature noises. Our method **differs fundamentally from adversarial or model inversion approaches**, which aim to generate visually interpretable concepts. Instead, the noises in Eq. (7)–(8) are designed to **modulate latent semantic directions** already encoded in the pretrained model—enhancing target-related features (Eq. 7) and diverging from them (Eq. 8).
>   - Following the insight of [1], which applies negative noise to “damage or overwrite the previously learned network weights for relevant classes”, we extend this idea by introducing paired positive–negative noises to **disentangle target-related and unrelated semantics**, achieving a balance between forgetting and retention. This disentanglement is controlled via Eq. (9).
>   - Theoretically, since the pretrained model is frozen, the noise $N$ minimizes the loss only when its perturbation aligns (Eq. 7) with the model’s class-level feature subspace, and maximizes it when misaligned (Eq. 8). Thus, $N$ captures semantically meaningful feature directions rather than arbitrary perturbations, ensuring concept-aware optimization without explicit constraints. When combined with target representations, these noises guide fine-tuning to reshape the model’s internal understanding of the target and perturbing concepts.
>   - Empirically, our ablation in Appendix F supports this claim: removing feature noises (“w.o.F.N”) disrupts the balance between forgetting and retention, leading to weaker unlearning and degraded generalization.
>   - [1] Tarun A K, Chundawat V S, Mandal M, et al. Fast yet effective machine unlearning[J]. IEEE Transactions on Neural Networks and Learning Systems, 2023, 35(9): 13046-13055.
> - W2&Q3: Extendibility of MaGA to other tasks
>   - We agree with the reviewer that evaluating concept-level unlearning on multimodal or generative models would be highly valuable. We believe the core idea of **semantic-guided, feature-level disentanglement, concept-level unlearning** can be potentially extended to LLMs, as MaGA operates directly in the semantic feature space, rather than relying on task-specific architectural assumptions. For instance, factual unlearning in LLMs could benefit from identifying and perturbing embeddings or captions associated with target data, guided by fact-similarity estimates. We consider this a promising direction for future work and will discuss this in the conclusion.
>   - **Extension to generative models.** Modern generative models (e.g., diffusion models) encode semantic concepts as structured latent directions. Since MaGA explicitly modulates and disentangles target-related latent directions in a pretrained model, the same mechanism can theoretically alter or suppress concept representations in generative settings.
>   - To further validate this, we conduct an additional experiment following prior work [2]. We reproduce CIFAR-10 concept-unlearning experiments using DDPM with classifier-free guidance. Two target classes (Airplane and Cat) are selected. Unlearning performance is evaluated using:
>     - UA, computed via an external ResNet-34 classifier to quantify forgetting;
>     - FID, computed on retained classes to assess generation quality preservation.
>   - As shown below, MaGA consistently achieves competitive UA and FID compared with both ESD and SalUn. Meanwhile, the performance of MaGA aligns most closely to that of the retrained model, indicating effective unlearning. Although due to time and computational constraints, we leave broader multimodal extensions to future work, these results already highlight MaGA’s strong potential to generalize across tasks.
> | class	| metric	| retrain	| ESD	|  SalUn	| MaGA  |
> |--------|--------|-------|-------|-------|-------|
> | Airplane | UA $\uparrow$		| 100.00	| 95.10	| 94.65	| 94.52	|
> | Airplane | FID $\downarrow$	| 12.86	| 18.25	| 12.03	| 12.41	|
> | Cat		| UA $\uparrow$		| 100.00	| 93.58	| 93.44	| 93.76	|
> | Cat 	| FID $\downarrow$	| 13.10	| 16.85	| 11.28	| 12.33	|
>   - [2] SalUn: Empowering Machine Unlearning via Gradient-based Weight Saliency in Both Image Classification and Generation

---

> ### Author Response · Authors · 2025-11-19
> **Author Rebuttal P2**
>
> - W3&Q2: The necessity of MLLM guidance
>   - We appreciate the reviewer’s question regarding the necessity of using an MLLM. Our rationale is threefold:
>   - **Semantic generalization:** MLLMs are trained on broad multimodal corpora and possess global conceptual knowledge, unlike classifiers or CLIP, which mainly encode dataset-specific or feature-level similarities. Since MLLM is capable of estimating semantic relationships in a zero-shot and dataset-agnostic manner, it is substantially more reliable to build the transition matrix encoding concept-level similarity.
>   - **Efficiency via caching:** Directly querying an MLLM for every forgetting sample would be expensive. Therefore, the transition matrix serves as a knowledge cache: MLLM similarities are computed once on a small exemplar subset and reused across all unlearning tasks, preventing MLLM from becoming a computational bottleneck.
>   - **Limitations of simpler alternatives:** Using classifier logits will lead to **overconfident and imbalanced perturbing labels**, producing unstable unlearning behavior (overfitting to dominant retained classes). On the other hand, CLIP focuses on feature similarity and lacks the global semantic reasoning needed for concept-aware perturbation.
>   - To better validate, we add an ablation extended from Appendix F to compare MLLM-based transition matrix, the CLIP-based transition matrix, and the classifier-based perturbing labels. Note that the M.T.M denotes “Model-inferred Transition Matrix”, and P.P.L denotes directly using pretrained model predictions for label assignment. MLLM consistently provides the best forgetting–retention tradeoff, validating its necessity.
> | class	| metric	| retrain	| M.T.M	| P.P.L	| CLIP	| MaGA	|
> |--------|--------|-------|-------|-------|-------|-------|
> | rocket	| $A_r$	| 76.19	| 70.96	| 71.26	| 74.63	| 75.75	|
> | rocket	| $A_f$	| 0.00	| 1.34	| 2.95	| 0.92	| 0.00	|
> | rocket	| MIA	| 8.06	| 15.70	| 11.80	| 1.40	| 0.00	|
> | MR    | $A_r$	| 76.23	| 69.21	| 69.61	| 74.78	| 76.25	|
> | MR    | $A_f$	| 0.00	| 1.18	| 2.60	| 0.00	| 0.00	|
> | MR    | MIA	| 6.01	| 18.90	| 15.84	| 5.55	| 0.00	|
> - W4: Further demonstration of experimental results.
>   - **Evaluation criterion and bolding rationale:** As defined in Sec. 5.1, effective unlearning requires: (i) the accuracies on forget data (A_f) and retain data (A_r) to be **as close as possible** to the retrained gold model, and (ii) a **low MIA score**, indicating reduced privacy risk and successful removal of target information. Following this criterion, in Tables 1-4, we bold the entries that best match the retrain model in both accuracy and MIA, rather than the numerically highest accuracy. We will revise the later version to explicitly report the gap from the retrain model for better clarity.
>   - **Consistency of improvements:** Although performance may vary slightly across classes, MaGA consistently achieves accuracies closest to retrain and lower MIA than baselines in most settings, which aligns with our definition of effective unlearning. This consistency across CIFAR-10/100 and multiple backbones on all unlearning tasks supports the robustness of our approach.
>   - We appreciate the reviewer pointing out the comparative baseline. SalUn [2] is designed primarily for a different unlearning setting (random forgetting and generative unlearning) and uses a distinct metric protocol from ours. Nonetheless, to ensure fairness, we **reproduced SalUn under our class-wise CIFAR-100 experimental setup** using ResNet-18 and added the results below. It is observed that MaGA achieves a better forgetting–retention tradeoff and lower MIA, confirming its competitiveness under the unified evaluation.
> | class	| metric	| retrain	| AMNC	| SSD	| SalUn	| MaGA	|
> |--------|--------|-------|-------|-------|-------|-------|
> | rocket	| $A_r$	| 76.19	| 73.59	| 75.86	| 75.70	| 75.75	|
> | rocket	| $A_f$	| 0.00	| 0.00	| 0.00	| 0.00	| 0.00	|
> | rocket	| MIA	| 8.06	| 28.62	| 0.66	| 0.00	| 0.00	|
> | MR    | $A_r$	| 76.23	| 73.22	| 76.20	| 76.00	| 76.25	|
> | MR    | $A_f$	| 0.00	| 0.00	| 0.00	| 0.00	| 0.00	|
> | MR    | MIA	| 5.61	| 46.06	| 0.25	| 0.75	| 0.00	|

---

> ### Author Response · Authors · 2025-11-25
> **Further Discussion**
>
> Dear Reviewer pJCC:
>
> We greatly appreciate your constructive comments, which have helped us improve this paper.
>
> We have carefully addressed all the concerns you raised, including the theoretical foundations of MaGA and the Fragment-Align strategy, the extendibility of MaGA to generative tasks, the necessity of MLLM guidance, and etc. We have also added further comparisons with the baselines you mentioned and provided additional clarification of the experimental results for better transparency.
>
> Having further discussions really helps to reach consensus and clarify misunderstandings. We would be grateful to hear any further concerns you may have, and we will do our utmost to address them thoroughly.
>
> Best,
>
> Authors.

---

### Official Review · Reviewer_yrJG · 2025-11-03

**Soundness:** 3
**Presentation:** 4
**Contribution:** 3
**Rating:** 6
**Confidence:** 4

**Summary:**

The authors introduce a novel unlearning method which focuses on addressing the forget-utility tradeoff for unlearning.

For a specific target image, their method first identifies other classes of images (called perturbing class) with similar feature representations as the target class, with the goal of mapping the representations of the target image to such similar classes.\
Then, the authors learn two noise matrices to add to the target image - a positive noise matrix which enhances the features common with the perturbing class and a negative noise matrix which enhances the features that are not representative of the target class.

This helps them to design a novel forget loss which maps the features of the target image to representations of a different class without damaging important features.

The authors demonstrate successful unlearning on a variety of unlearning settings with significant improvement in MIA accuracy in many cases.

**Strengths:**

- The paper is well written and easy to follow.
- The authors introduce a novel algorithm with a focus on preserving model utility.
- The authors provide comprehensive experiments on CIFAR-10, CIFAR-100 and CIFAR-20 using Resnet18 and ViT.

**Weaknesses:**

The main weakness is some missing baselines.
Kindly compare with other SOTA methods like [1] and [2].
Even though [2] is designed for class-wise unlearning, it is designed using a similar rationale of class-discriminatory feature space manipulation.



[1] Kurmanji, Meghdad, et al. "Towards unbounded machine unlearning." Advances in neural information processing systems 36 (2023): 1957-1987.\
[2] Kodge, Sangamesh, Gobinda Saha, and Kaushik Roy. "Deep unlearning: Fast and efficient gradient-free class forgetting." Transactions on Machine Learning Research (2024).

**Questions:**

- What happens if the model logits are used to form the transition matrix instead of the MLLM ?
- Is there any rationale behind the design choice of a True / False question for the prompt to the MLLM ?
- In Figure 1,  please indicate which one is the image from the target class and perturbed class. Are these masks found using optimization ?  If not, kindly include some examples of positive and negative noise matrices.
- Please specify the function $\phi$ in Eq. 6
- Is there any constraint on the feature noise matrices ?
- Additionally, it is unclear what the optimizations of Eq 7 and 8 are doing.  The subscript i of $N_i$ indicates that it is specific to $x_i$, however that does not seem to be the case from the optimization itself. Please give an intuition as to why only $N$ is used rather than $N+x_i$ inside the loss functions.

---

> ### Author Response · Authors · 2025-11-19
> **Author Rebuttal**
>
> Thanks for the thoughtful and encouraging feedback.
> - W1: About baselines.
>   - We appreciate the reviewer’s suggestion regarding missing baselines. We will add discussions on [1] and [2] in Section 2.
>   - [1] adopts a conditional teacher–student framework for selective knowledge transfer.
>   - [2] follows a model-pruning-like intuition, estimating class-specific subspaces via SVD to remove target-class-discriminatory responses. In contrast, our method fine-tunes the model with positive and negative feature noises to re-align overlapping semantics of target data with retained data instead of removing them, enabling more complete unlearning and better generalization to multi-class or random forgetting.
> - Q1: Using model logits for the transition matrix.
>   - The **major limitation of using model logits** for the transition matrix is that it will lead to overconfident and biased predictions, producing imbalanced transition matrices and dominant perturbing labels, which harm unlearning stability with a risk of model overfitting to certain dominant classes during unlearning.
>   - Ablations on CIFAR-100 (ResNet18) show that MLLM-derived transitions achieve lower forget accuracy, comparable retain accuracy, and significantly lower MIA risk than model-logit-derived ones (M.T.M denotes “Model-inferred Transition Matrix”):
> | class  | metric | retrain | M.T.M | MaGA  |
> |--------|--------|-------|-------|-------|
> | rocket | $A_r$    | 76.19 | 70.96 | 75.75 |
> | rocket | $A_f$    | 0.00 | 1.34  | 0.00  |
> | rocket | MIA    | 8.06 | 15.70 | 0.00  |
> | MR     | $A_r$    | 76.23 | 69.21 | 76.25 |
> | MR     | $A_f$    | 0.00 | 1.18  | 0.00  |
> | MR     | MIA    | 6.01 | 18.90 | 0.00  |
> - Q2: MLLM prompt designs.
>   - We use a True/False format to stabilize MLLM outputs via In-Context Learning. Direct multi-class prompting often causes inconsistent logits; pairwise binary queries encourage stable semantic similarity estimation between two concepts.
>   - Although prompt order slightly affects absolute logits, we rely only on **relative inter-concept similarities** for constructing the transition matrix, so this variation has no impact.
> - Q3: About Figure 1.
>   - Figure 1 serves as an illustration rather than a visualization of actual samples. It is intended to intuitively demonstrate the overlap of feature representations across different semantic concepts, which motivates our later disentanglement strategy.
>   - In this example, if “dinosaur” is the target concept to be forgotten, the goal of our unlearning is to suppress its unique features (marked in green) while preserving the associated features (marked in red) that are shared with other classes, ensuring that the forgetting process does not damage generalizable representations.
>   - We will add visual examples of positive and negative feature noises in the revision to illustrate this semantic coupling.
> - Q4: The specification of Equation 6.
>   - The function $\phi$ in Eq. 6 denotes the **ranking and selection operation** over the transition probability vector $R_i$ obtained in Eq. 5. Specifically, $R_i$ encodes the feature-level semantic similarities between the current target instance and all candidate concepts. The function $\phi$ first sorts the elements of $R_i$ in descending order and then selects the class whose similarity ranks at the top -( \tau \times 100% ) as the perturbing label $y_{i}^{p}$.
> - Q5: The constraints on the feature noise matrices.
>   - The feature noises are constrained by **L2 normalization** to prevent large magnitudes.
>   - Moreover, the two noise matrices differ in their optimization objectives: the positive noise $N_{i}^{pos}$ is optimized toward the perturbing label to inject features semantically aligned with the alternative concept, guiding the model to re-anchor the target representation; the negative noise $N_{i}^{neg}$, in contrast, is optimized in the opposite direction to damage and overwrite model responses associated with the ground-truth label.
> - Q6: The optimization of feature noises.
>   - In our framework, feature noises are generated **per class**, not per instance. Therefore, Eq. 7–8 optimize only with respect to the class label $y_{i}$ (or $y_{i}^{p}$), not $x_i$. The notation $(x_{i}, y_{i})$ in Line 176 merely indicates that each target sample later uses its class-specific noise. We will clarify this in the revised version to avoid misunderstanding.
>   - Intuitively, since the pretrained model already captures semantic representations of each class, when we freeze it and optimize $N$ toward a given label $y$, the noise progressively learns to approximate feature directions that belong to (for Eq. 7) or diverge from (for Eq. 8) that class’s latent representations, thereby yielding the distinct effects discussed in Sec. 4.2.

---

> ### Author Response · Authors · 2025-11-25
> **Further Discussion**
>
> Dear Reviewer yrJG:
>
> We want to express our appreciation for your valuable suggestions, which greatly helped us improve the quality of this paper. We are also grateful that you found our idea novel and the paper well-written and easy to follow, supported by comprehensive experiments.
>
> We have made our maximum effort to address all your concerns, including the alternative construction of transition matrix, the MLLM prompt design, the clarification of Eq. (6), the theoretical basis and constraints of feature noise optimization, and etc. We have also added a discussion of the related baselines you mentioned, which will be further integrated into the next full revision of the paper.
>
> Your continued feedback is important for evaluating the improvements in our revised manuscript, and we sincerely look forward to hearing your further thoughts. Thank you again for your time and support.
>
> Best,
>
> Authors.

---

### Author Response · Authors · 2025-11-19
**General Author Rebuttal**

We sincerely thank all reviewers for their time and insightful comments. We appreciate the positive feedback, including:
- “The paper is well written and easy to follow.” (R1)
- “The use of semantic concept modeling is insightful and could inspire follow-up research on concept-aware unlearning” (R2)
- “Novel conceptual direction…” (R3)
- “…provide comprehensive experiments…”(R1)

We also appreciate the reviewers’ constructive suggestions. Below, we summarize our clarifications and part of the newly added experiments.
- R2 and R3 encouraged extending our experiments to provide further insights into MaGA.
  - We expand the discussion of the intuition and theoretical basis of MaGA, highlighting its potential to generalize beyond classification to LLM-based or generative-model unlearning.
  - Following prior work [1], we add experiments on **image generation unlearning** using DDPM with classifier-free guidance. Results show that MaGA remains competitive and closely aligned with retraining, demonstrating transferability beyond classification.
- R1, R2, andR3 questioned the role of feature noises in semantic disentanglement. We strengthened our explanation and provided new ablations.
  - Theoretical clarification
    - With the backbone frozen, the noise $N$ can only reduce loss when its perturbation aligns with the class-level latent semantics (Eq. 7) and increase it when misaligned (Eq. 8). Thus, the paired noises effectively **modulate latent semantic directions** already encoded in the pretrained model.
    - MaGA reshapes the pretrained feature space: negative noise disrupts the target distribution, while positive noise reassigns representations to new semantic regions. They cooperatively realign the decision boundary in a direction guided by the MLLM (perturbing labels).
  - We add ablations comparing “w.P.F”(with positive feature), “w.N.F”(with negative feature), and MaGA. It is observed that neither variant is sufficient. Only the **paired noises** achieve the disentanglement necessary for strong unlearning and retention.
| class	| metric	| retrain	| w.o.F.N	| w.P.F	| w.N.F	| MaGA	|
|--------|--------|-------|-------|-------|-------|-------|
| rocket	| $A_r$	| 76.19	| 75.46	| 75.91	| 74.63	| 75.75	|
| rocket	| $A_f$	| 0.00	| 3.32	| 10.35	| 3.44	| 0.00	|
| rocket	| MIA	| 8.06	| 0.00	| 8.40	| 0.00	| 0.00	|
| MR    | $A_r$	| 76.23	| 76.07	| 76.22	| 74.78	| 76.25	|
| MR    | $A_f$	| 0.00	| 4.55	| 9.50	| 0.00	| 0.00	|
| MR    | MIA	| 6.01	| 0.00	| 10.20	| 1.68	| 0.00	|
- R1 and R2 questioned the necessity of MLLM guidance and asked whether alternative perturbing-label strategies (e.g., logits, CLIP) suffice.
  - **MLLM guidance is essential in MaGA** because the perturbing label must be:
    - **Semantically consistent:** to guide meaningful realignment toward related concepts;
    - **But incorrect:** to ensure not too close to the original label to enable effective forgetting, controlled via the ranking threshold $\tau$.
  - We add experiments comparing MLLM-based transition matrix, CLIP-based transition matrix, and classifier-based perturbing labels to demonstrate MLLM’s superiority in our framework.
| class	| metric	| retrain	| M.T.M	| P.P.L	| CLIP	| MaGA	|
|--------|--------|-------|-------|-------|-------|-------|
| rocket	| $A_r$	| 76.19	| 70.96	| 71.26	| 74.63	| 75.75	|
| rocket	| $A_f$	| 0.00	| 1.34	| 2.95	| 0.92	| 0.00	|
| rocket	| MIA	| 8.06	| 15.70	| 11.80	| 1.40	| 0.00	|
| MR    | $A_r$	| 76.23	| 69.21	| 69.61	| 74.78	| 76.25	|
| MR    | $A_f$	| 0.00	| 1.18	| 2.60	| 0.00	| 0.00	|
| MR    | MIA	| 6.01	| 18.90	| 15.84	| 5.55	| 0.00	|
- R3 raised concerns about the computation cost of MLLM.
  - We emphasize that MLLM inference is **not a bottleneck**, due to the following reasons:
    - MLLM is queried only once per dataset to estimate inter-class semantic similarity in a zero-shot fashion. No training or fine-tuning is required, keeping the cost minimal
    - We use a transition matrix to cache the MLLM knowledge for perturbing labels, which is computed using only 3–10 exemplar images per class.
    - This process is fully **decoupled from the training or unlearning process** and can be pre-computed once per dataset, **even before model training**, and reused for all subsequent unlearning tasks. Thus, its cost is amortized and negligible for all practical unlearning tasks.


[1] Fan C, Liu J, Zhang Y, et al. Salun: Empowering machine unlearning via gradient-based weight saliency in both image classification and generation[J]. arXiv preprint arXiv:2310.12508, 2023.

---

### Author Response · Authors · 2025-12-01
**Rebuttal Summary**

Dear Reviewers, AC, and SAC,

We sincerely thank the AC and SAC for their dedicated efforts, and we deeply appreciate the reviewers for taking the time to evaluate our work and provide constructive feedback. During the discussion phase, we are glad to see that one reviewer confirmed that **most concerns had been resolved and raised the score**.

Although the remaining reviewers did not participate further in the discussion, we believe that we have made our utmost efforts to thoroughly address their concerns through both theoretical analysis and empirical validation, including:
- Additional unlearning experiments on image generation tasks, providing deeper insights into MaGA and demonstrating its generalizability across diverse unlearning scenarios.
- Further clarification of the role of feature noises and a strengthened theoretical explanation of its optimization. Additional ablation studies were included to support these clarifications.
- A more detailed explanation of the necessity of MLLM guidance, showing that MLLMs help assign the **most appropriate** perturbing labels to manipulate the model re-optimization direction. We further compared alternative label-assignment strategies to highlight the effectiveness.
- Clearer specification of the computation of the transition matrix and feature noises, demonstrating that these components do not introduce significant computational overhead.

Since further interactive discussion was not possible, we would like to re-emphasize the **key contributions and significance** of our work:
- We introduce **a novel concept-aware unlearning algorithm**, leveraging MLLM-guided re-optimization and a feature-disentanglement strategy to address the trade-off between over-unlearning and insufficient unlearning.
- We conduct comprehensive experiments across multiple unlearning tasks and datasets to thoroughly assess the proposed method, which demonstrates its potential to serve as a foundational approach for broader machine learning security settings, such as privacy-driven data removal and controlled modification or correction of model knowledge.
- We provide a clear and detailed workflow of our algorithm, which reviewers recognized as “inspiring for follow-up research” (R2) and “easy to follow” (R1).

We have devoted extensive effort over the past several months to refining this work, and we remain committed to further improving clarity and presentation according to the reviewers’ suggestions. We hope that our work will contribute meaningful insights to the community and advance the study  of machine unlearning.

Thank you once again for your thoughtful feedback and valuable time.

Sincerely,

The Authors

---

### Meta-Review · Area_Chair_nLXj · 2026-01-07

**Summary:**

The main concerns (in my opinion) raised by the reviewers were the following:
- The proposed scheme has heavy reliance on a MLLM (multi-modal large language model) to generate the "transition matrix" which begs the following questions:
  - It is not clear how the method would perform when we change the MLLM or its capabilities.
  - On a related note, it appears that the form of the MLLM prompt is quite important, and these prompts are critical to creating the necessary transition matrix. It is not clear how robust (or fragile) the proposed scheme is to changes to the exact prompt itself (authors indicate that they had to create specific form of true/false prompts); heavy reliance on hand-crafted prompts make the scheme too fragile.
  - It is not clear the how the method would perform if the images/features or the class labels are not semantically meaningful (think class labels "A", "B", "C", etc) or the images are from domain not part of the MLLM training distribution (for example, the images are X-ray images, and the MLLM is not trained on xray images at all).
- Lack of technical motivation for the proposed scheme, making the proposed scheme somewhat unintuitive, and the author response doesn't provide adequate justification in my opinion:
  - While the proposed scheme shows strong empirical performance, being the closest to the retrain baseline metrics in many cases, it is not clear why that should happen given the proposed scheme. Some intution at a technical level is necessary to understand why the process used in this submission pushes us towards the retrain baseline.
  - On a related note, usually the goal of approximate unlearning is the match the metrics of the retrain models across varying metrics. However, in case of MIA (membership inference attacks), the MIA scores of the proposed scheme is significantly lower than that of the retrained model, which highlights a divergence from the "gold-standard model". This needs proper discussion as to why this is desirable.
  - Another technical issue from the standpoint of unlearning is that if the example to be unlearned is part of the sample utilized to generate the transition matrix (or even part of the training data of the MLLM used), it is not clear how that would be handled in this proposed scheme.

For these reasons, I am suggesting a reject for this submission.

**Reviewer Concerns:**

Beyond the main concerns discussed in the **Summary** section, the following concerns were discussed:

- Multiple reviewers brought up the potential computational cost of MLLM inference in the unlearning process.
  - The authors adequately addressed this issue by highlighting that this MLLM inference is a onetime cost amortized over the training and all subsequent unlearning requests.
  - However, this one-time fixed transition matrix conflicts with the aforementioned situation where the unlearned example was used to generate the transition matrix. From an unlearning perspective, it makes sense that the transition matrix is appropriately updated as examples are unlearned if they were used in forming the transition matrix.
- Proper positioning and comparison with existing similar schemes and additional unlearning problems.
  - The reviewers mentioned that the unlearning problems considered in the submission were quite limited. In the author response, the authors addressed this by providing new results on unlearning in an image generation problem.
  - The reviewers mentioned that critical ablations for the proposed paired-noise, and the authors provided results demonstrating that the paired noise is necessary for the best utility-unlearning tradeoff.
  - The reviewers questioned the reliance on a MLLM, and suggested alternate schemes for generating the transition matrix. The author response presented results showing that the MLLM based transition matrix provides better performance, though, in some cases, the CLIP based transition matrix led to performance closer to the retrained baseline.
  - The reviewers brought up relevant baselines such as SalUn, which the authors compared against in their response, demonstrating that the proposed scheme is competitive to these baselines.

**Reviewer Scores:**

Reviewer Xuxu initially scored the submission as a 2, and while they mentioned increasing their score after the author response, the reviewer still seems unconvinced regarding the acceptability of the submission, and their assessment seems reasonable to me.

Reviewer yrJG initially scored the submission as a 6, and I believe their concerns would be partially addressed so they might raise their scores by at most 1.

Reviewer pJCC initially scored the submission as a 4, and I do not think they would raise their scores as I believe that their concerns regarding reliance on a MLLM and the lack of technical motivation was adequately addressed.

---

### Decision · Program_Chairs · 2026-01-26

Reject